# Remote Sensing Grassland Productivity Attributes: A Systematic Review

**Tsitsi Bangira** [1,*], **Onisimo Mutanga** [2], **Mbulisi Sibanda** [3], **Timothy Dube** [4]
**and Tafadzwanashe Mabhaudhi** [1,5]

1   Centre for Transformative Agricultural and Food Systems, School of Agricultural, Earth and Environmental Sciences, University of KwaZulu-Natal (UKZN), Scottsville, Pietermaritzburg 3209, South Africa; mabhaudhi@ukzn.ac.za
2   Discipline of Geography and Environmental Science, School of Agricultural Earth and Environmental Sciences, University of KwaZulu-Natal, Private Bag X01, Scottsville, Pietermaritzburg 3209, South Africa
3   Department of Geography, Environmental Studies & Tourism, Faculty of Arts, University of the Western Cape, Bellville 7535, South Africa
4   Institute of Water Studies, Department of Earth Sciences, University of the Western Cape, Private Bag X17, Bellville 7535, South Africa
5   International Water Management Institute (IWMI), Pretoria 0127, South Africa
*   Correspondence: bangirat@ukzn.ac.za

**Abstract:** A third of the land on the Earth is composed of grasslands, mainly used for forage. Much effort is being conducted to develop tools to estimate grassland productivity (GP) at different extents, concentrating on spatial and seasonal variability pertaining to climate change. GP is a reliable indicator of how well an ecosystem works because of its close connection to the ecological system equilibrium. The most commonly used proxies of GP in ecological studies are aboveground biomass (AGB), leaf area index (LAI), canopy storage capacity (CSC), and chlorophyll and nitrogen content. Grassland science gains much information from the capacity of remote sensing (RS) techniques to calculate GP proxies. An overview of the studies on RS-based GP prediction techniques and a discussion of current matters determining GP monitoring are critical for improving future GP prediction performance. A systematic review of articles published between 1970 and October 2021 (203 peer-reviewed articles from Web of Science, Scopus, and DirectScience databases) showed a trend in the choice of the sensors, and the approaches to use are largely dependent on the extent of monitoring and assessment. Notably, all the reviewed articles demonstrate the growing demand for high-resolution sensors, such as hyperspectral scanners and computationally efficient image-processing techniques for the high prediction accuracy of GP at various scales of application. Further research is required to attract the synthesis of optical and radar data, multi-sensor data, and the selection of appropriate techniques for GP prediction at different scales. Mastering and listing major uncertainties associated with different algorithms for the GP prediction and pledging to reduce these errors are critical.

**Keywords:** grassland ecosystem services; LAI; aboveground biomass; canopy storage capacity; chlorophyll and nitrogen content

## 1. Introduction

Grasslands, covering at least one-third of the Earth's land surface, provide different ecosystem services, including carbon sequestration, biodiversity conservation, forage, and opportunities for tourism and recreation [1–4]. From a climate change perspective, grasslands, both in tropical and temperate regions, play a significant function in maintaining the carbon (C) cycle and balancing greenhouse gases (GHGs) [3,5]. These ecosystems contribute roughly 12% of the total terrestrial carbon stocks, and any changes in their quality and quantity can potentially change their role in the C cycle [5,6].

Over 20% of the world's grasslands appear to be threatened, and more than 7.5% of them appear to be disturbed [7]. Over the past ten years, grassland degradation has been estimated to have cost the global livestock industry more than USD 7 billion. The impact on socioeconomic life is particularly alarming in underdeveloped areas, where most communities depend on grasslands for feeding livestock [8]. As a result, grassland degradation portrays a critical problem that has to be addressed to maximize their potential to provide ecosystem services in the future.

To accurately assess grassland ecological status, certain traits and indicators need to be investigated [9]. Indicators are measurable parameters that can be used to assess the current state of key ecological attributes and provide warnings about potential threats to the ecosystem [10]. Grass quantity and quality are indicators of grassland productivity (GP), management practices, and the ecological processes that affect them. The quantitative and qualitative traits of GP monitoring include aboveground biomass (AGB), yield, leaf area index (LAI), canopy storage capacity (CSC), and photosynthetic activity. Recently, RS has been widely used to acquire information on the quality and quantity of grasslands over large areas, and is relatively cheaper than conventional field surveys [11,12]. These quantitative and qualitative proxies are frequently used, equally, for GP monitoring in RS studies. As a result, these grassland output monitoring attributes and indicators are also evaluated in this context. There have been several successful studies on the prospective and capability of RS systems for GP monitoring [13–15]. For example, Naidoo et al. [16] utilized random forest regression models, derived from WorldView space-borne sensors (WV3), to yield the highest AGB prediction accuracies (RMSE = 169.28 g/m$^2$). Similarly, Guerini Filho, Kuplich, and Quadros [15] made use of Sentinel-2 data to predict biomass in the Brazilian Pampa using a multiple linear regression analyses approach and produced high modelling accuracies ($R^2 > 0.8$). In another study, Quan et al. [17] compared radiative transfer model (RTM) AGB estimation to those obtained using an exponential regression, a partial least square regression (PLSR), and artificial neural networks (ANNs). The RTM-based method (RMSE = 41.65 gm$^{-2}$) performed better than the exponential regression (RMSE = 42.67 gm$^{-2}$) and the ANN (RMSE = 46.26 gm$^{-2}$). However, to date, RS-based approaches for predicting proxies for GP are still in their infancy, not used extensively, and frequently implemented with undetermined accuracy [18–20]. There is a dearth of consistent approaches, particularly for grassland ecosystems [21]. Robust and transferable techniques to estimate proxies for GP are still needed. Furthermore, the introduction of new and more sophisticated sensors demonstrates that RS data will continue to contribute significantly to studies on GP estimation [22–24].

This study presents a comprehensive systematic review of the scientific peer-reviewed articles on using remotely sensed data within the explicit theme of estimating GP proxies, such as AGB, LAI, CSC, yield, and chlorophyll content. The study presents examples from the literature that summarize the remote sensing of the GP landscape, chronicling the evolution of sensors and associated methodologies, and analyzing the geographic distribution of studies at various spatial scales. In this instance, specific search terms were used to locate information on the RS platform used, the characteristics of the sensor used, the extent of the study, and the prediction accuracy of the algorithms used.

## 2. Materials and Methods

The phrases and definitions of grasslands differ from study to study, according to its scope and objectives. The tag, grasslands, is used to symbolize unidentified graminoids [25] or as a broad term for different kinds of grasses (e.g., pasture lands) [26,27]. Terms such as rangelands, pasturelands, or savanna can be found in the pertinent literature [4] to represent the land use activities of grasslands, whether natural or man-made. This study assessed natural grasslands that provide regulatory and provisioning ecosystem services.

This review used a systematic approach [28] to identify peer-reviewed articles that published original research on estimating GP proxies using RS. This review is divided into two sections to accomplish its research objectives. The initial section focuses on the

growth achieved, to date, in estimating and monitoring GP using remotely sensed data. The biophysical variable parameters, sensor characteristics, sensor platforms, approaches, and suitable spectral characteristics that have been utilized to date are reported in this section. The last section highlights the recommendations and the way forward for future studies focusing on GP. The detailed literature search and analysis were conducted in four stages:

1. Stage 1: Literature search

The preliminary stage of the relevant articles search was to compile a list of all key texts, words, phrases, and terms found in search strings. These terms must appear in the article title and keywords of the abstract. The preliminary literature was searched in Google Scholar, and the few top articles were downloaded and analyzed for keywords. The following texts and their alternatives were used: "grassland productivity", "remote sensing", "GIS", "grassland productivity monitoring'', "above ground biomass", "leaf area index", "yield", "grassland nutrients AND Remote Sensing & GIS", "grassland productivity AND remote sensing", "grassland productivity monitoring", "grassland above ground biomass AND Remote sensing & GIS", "grassland LAI and Remote sensing & GIS", "grassland canopy storage capacity and remote sensing", and "grassland productivity AND yield AND remote sensing & GIS." In some searches, the word grassland was replaced with any of the following terms: prairie*, meadow*, pasture*, savanna*, veld*, steppe*, 'old field'*, and shrub*. The inclusion/exclusion criteria were restricted to title, abstract, and keywords. Table 1 shows the query strings used across the databases.

**Table 1.** Key terms utilized in the literature search.

| Database | Search Strings | Studies Reserved |
|---|---|---|
| SCOPUS | TITLE-ABS-KEY (("grassland") AND ("remote sensing") AND ("aboveground biomass")) OR ("leaf area index") OR (("grass chlorophyll content ") AND ("remote sensing")) OR (("grassland yield") AND ("remote sensing") & AND GIS)) OR (("grassland quality") AND ("remote sensing" & GIS)) OR ((grassland nitrogen) ("remote sensing" & AND GIS)) OR (("grassland canopy storage capacity") OR ("remote sensing")) OR (("grassland productivity") AND ("remote sensing" & GIS)) AND (LIMIT-TO (LANGUAGE, "English")) | 1403 |
| ScienceDirect | "grassland" OR "remote sensing" AND "grassland chlorophyll content" AND "grassland canopy water storage" OR "grassland aboveground biomass" OR "yield" OR "grassland quality" AND "Remote sensing & GIS" OR "leaf area index" | 869 |
| Web of Science | TS = (("grassland") AND ("remote sensing" OR "GIS") OR (grassland "leaf area index") OR ("canopy storage capacity") OR (grassland "aboveground biomass") OR ("grassland quality")) | 2348 |
| Google Scholar | No key terms were used. Articles from the reference list. | 135 |
| | Full-text articles assessed for eligibility | 1289 |
| | Articles | 203 |

The identified keywords were pasted in the SCOPUS, ScienceDirect, and the Web of Science databases to build the relevant literature database. The missing papers from Web of Science, Scopus, and ScienceDirect were located using Google Scholar. Pertinent articles were also found in the list of references of the relevant studies through a reverse reference list inspection [28]. The literature was further screened and filtered to ensure that the primary focus of the review was the RS of grassland productivity. Peer-reviewed articles published between 1975 and the end of November 2021 were considered.

2. Stage 2: Screening

Preliminary literature searches in SCOPUS, ScienceDirect, Web of Science, and Google Scholar yielded 1403, 2348, 869, and 135 studies, respectively (Figure 1 and Table 1). In preparation for screening, the abstracts and keywords from the retrieved papers were exported to Endnote. The initial screening procedure included the removal of duplicate articles as well as those written in languages other than English. The next step involved a comprehensive examination of the articles based on the use of RS to estimate AGB, LAI, and grass nutrient and chlorophyll content as proxies of GP. Only full-length articles of the chosen abstracts were included for detailed analysis. The detailed information for each article was then captured in a spreadsheet. Two hundred and three articles were considered for the quantitative and qualitative analysis of this review.

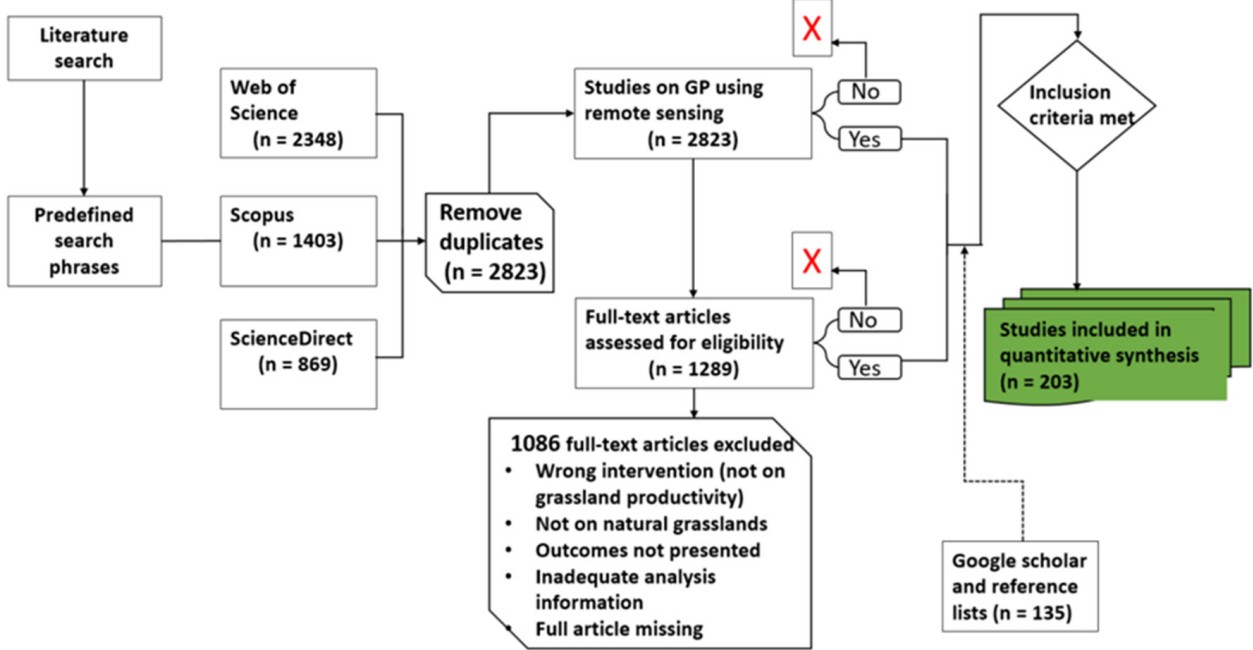

**Figure 1.** Steps followed for the articles considered in this review.

3. Stage 3: Data retrieval

All articles retained from the preceding stage were used to comprehensively indicate the current development, gaps, problems, and strengths in using RS techniques to estimate GP. To answer the research questions of this study, the third stage extracted data from the identified articles. The data recorded included details on the year the research was conducted, the spatial extent of the study site, the sensor attributes (such as spatial, temporal, radiometric, and spectral resolution), the proxy used to estimate GP, vegetation indices used, prediction accuracies, and the algorithms used. The explicit attributes were then changed to measurable variables, in preparation for the data analysis phase. Concurrently, key bibliometric data, including author names, country, publication year, article title, and abstract, were also gathered during this stage. Missing studies not identified in previous

stages were captured during this stage. Accordingly, the stage evaluated the systematic review's applicability and quality assessment stage.

4.     Stage 4: Data analysis

The literature that was gathered and extracted was analyzed both qualitatively and quantitatively. Regarding the quantitative analysis of studies focusing on the quantification of proxies of GP, a principal statistical analysis, such as trends, progress, and future projections, was conducted. A bibliometric survey was also performed to disclose the trends of principal words and phrases in monitoring GP. In addition, this aided in identifying research hotspots, development trends, the most cited authors, most relevant publications, and most frequently utilized keywords within a research area [29,30].

VOSviewer software developed by van Eck and Waltman [31] was used for text mining and presenting bibliometric maps of key terms used to estimate and monitor GP. The titles, keywords and abstracts of the studies in the resultant database (203 articles) were entered into VOSviewer to investigate GP. Literature analyses can be biased, but considering that only the existence and co-existence of important texts and frequency distributions were evaluated, a bias evaluation was not prepared. To avoid biased reporting, the PRISMA (http://www.prisma-statement.org/, accessed on 26 August 2022) statement was used as a guide [32].

The study area of each article was evaluated in terms of country and continent. The spatial scale of the analysis was also considered. The scale of the studies was grouped into five categories, namely, local (<1000 km$^2$), landscape ($\geq$1000 km$^2$), national (entire country), regional (multiple countries), and global. If the study included multiple countries, all the countries were listed.

The detailed analysis of data usage included an extensive assessment of the different RS systems used in GP monitoring. Both microwave and optical sensors were considered in this review. These sensors can be on board either satellite, ground, or airborne platforms. The category of reference data used for the accuracy assessments was also identified.

## 3. Results

### 3.1. Searched Literature Traits: Published Trends

The first publications for GP monitoring using remote sensing were made in 1976 [33,34], considering that the first Earth-observing satellite launched to monitor and study terrestrial ecosystems became functional in 1972. Since then, the number of studies has increased steadily, with a significant number of articles published by the end of 2021 (Figure 2). The first steady increase in publication activity occurred in the early 2000s. Between 2010 and 2018, a span of eight years, the annual publication rate increased to at least one article per month on average. The period of 2018–2020 is clearly noticeable since the publication amounts nearly doubled this period in contrast to the previous era.

### 3.2. Keyword Analysis

Figure 3 depicts the development and direction of research based on key terms taken from the published paper titles, abstracts, and keywords used in this study. Text mining makes it possible to discover the development of the occurrence of research phrases in the analysis period. Three clusters in purple, green, and yellow across the past four decades are visible. The purple cluster has the most centralized, historical terms and is characterized by words such as "avhrr", "ndvi", "savi", "wavelength", "biophysical parameter", "nitrogen", and "phytomass". The co-occurrence of the coarse resolution AVHRR sensor and regional study sites, such as China and the Tibetan Plateau, implies the preference for a coarse resolution sensor when the study area is large.

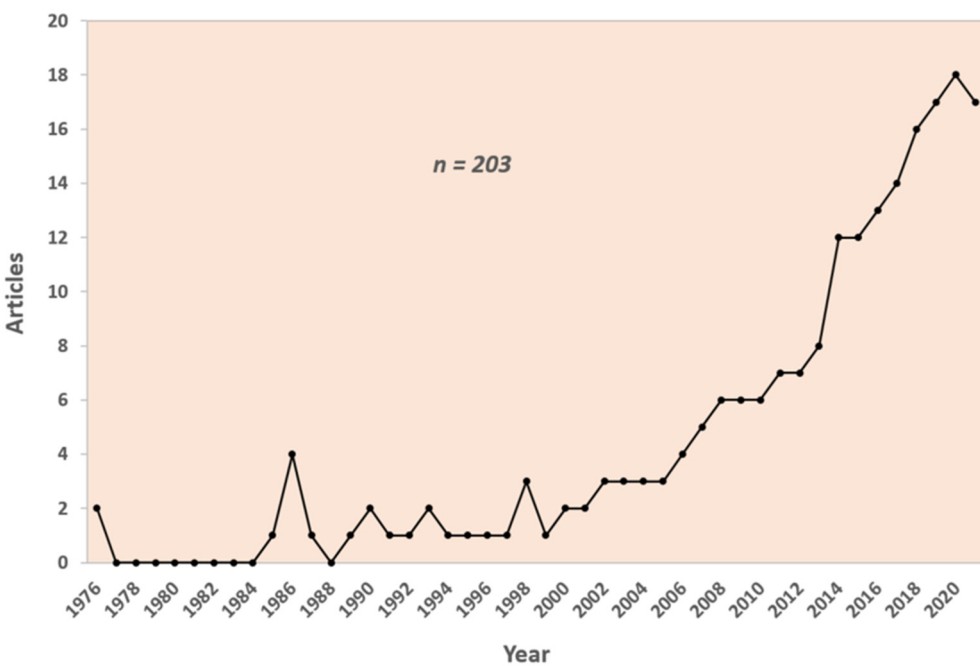

**Figure 2.** Evolution in the time of the published articles that used remotely sensed data for GP monitoring.

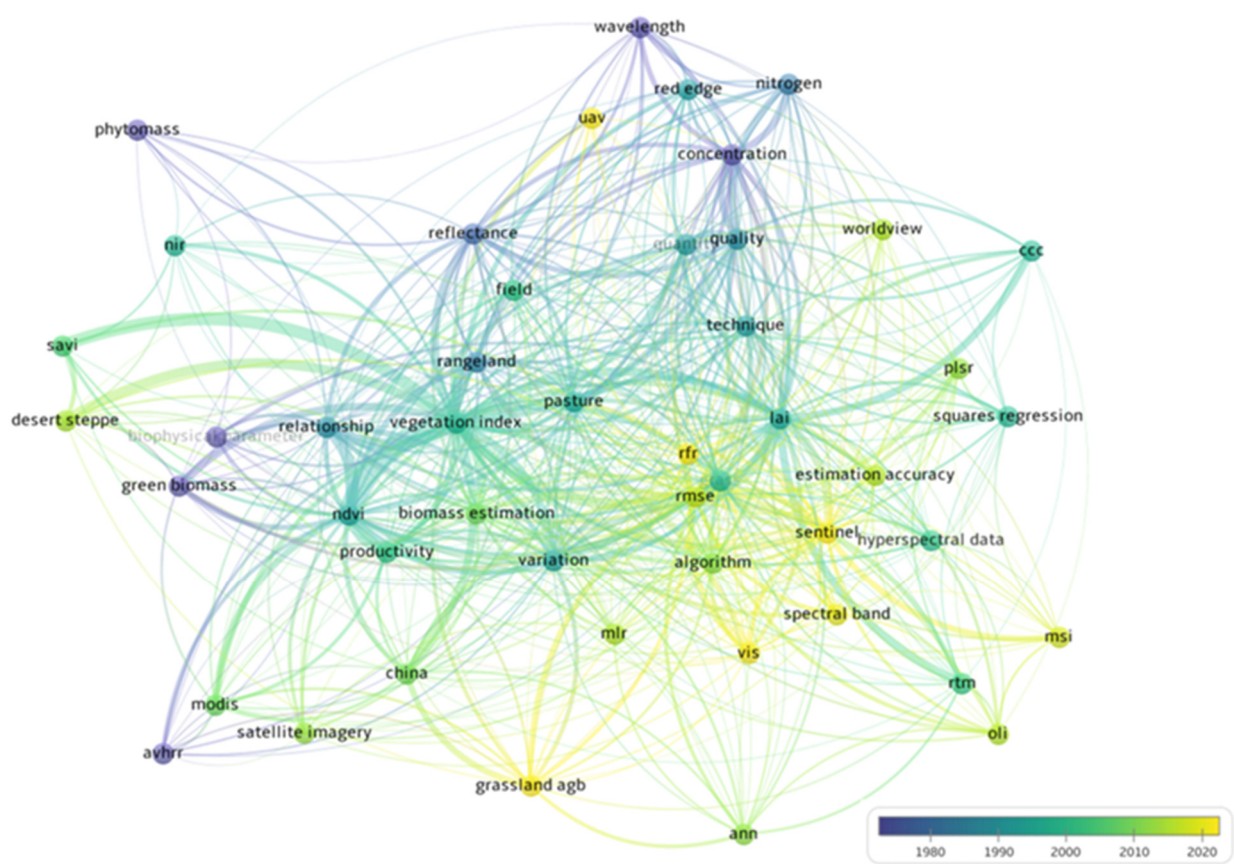

**Figure 3.** Evolution of key terms in estimating grassland productivity using remote sensing, based on the data derived from the abstracts and titles of the selected literature.

The green cluster, which covers the years from 1995 to 2005, has the terms "vegetation index", "leaf area index (lai)", "canopy chlorophyll content (ccc)", "radiative transfer model

(rtm)", "MODIS", "artificial neural network (ann)", "vegetation index)", "landsat", "red-edge", "quality", and "quantity". The studies conducted during this period were more based on the indicators of forage quantity (e.g., AGB and LAI) [35–37] rather than quality (canopy chlorophyll content and nutrients) parameters.

The yellow cluster showed "sentinel", "operational land imager (oli)", "machine learning regression (mlr)", "random forest regression (rfr)", "worldview", "unmanned aerial vehicle (uav)", and "PLSR". This indicates a noticeable shift from conventional classification techniques to more robust MLAs, such as the partial least squares regression (PLSR) and random forest (RF) ensembles, in predicting proxies of GP. The trend in the sensors illustrates the peak in the utilization of AVHRR and MODIS in the historical studies published before 2015. This trend shifted towards the recently launched Landsat-8 OLI (launched February 2013) and Sentinel-2 (S-2) MSI (launched June 2015) instruments associated with improvements in estimation algorithms.

It is worth noting that many of the studies used AGB, LAI, and chlorophyll content as proxies for evaluating and monitoring GP. Few studies used the measurement of grassland quality traits, such as nutrient content, as indicators of GP.

### 3.3. Geographic Patterns

Figure 4 shows that the geographical distribution of the articles considered in this study is uneven across all continents. More research has been conducted in Asia, particularly in China's "Tibetan Plateau" and "Inner Mongolia". In particular, China has 54 articles relating to GP, followed by the USA, with 38 published articles. Canada, Italy, and Brazil are the other countries where considerable studies on GP have been conducted. Australia has conducted the least amount of studies, followed by South America. Although considerable research has been conducted in Africa, many articles were conducted in South Africa. It is noteworthy that only eight studies were conducted across multiple countries. This shows that significant efforts are required for widespread GP surveillance.

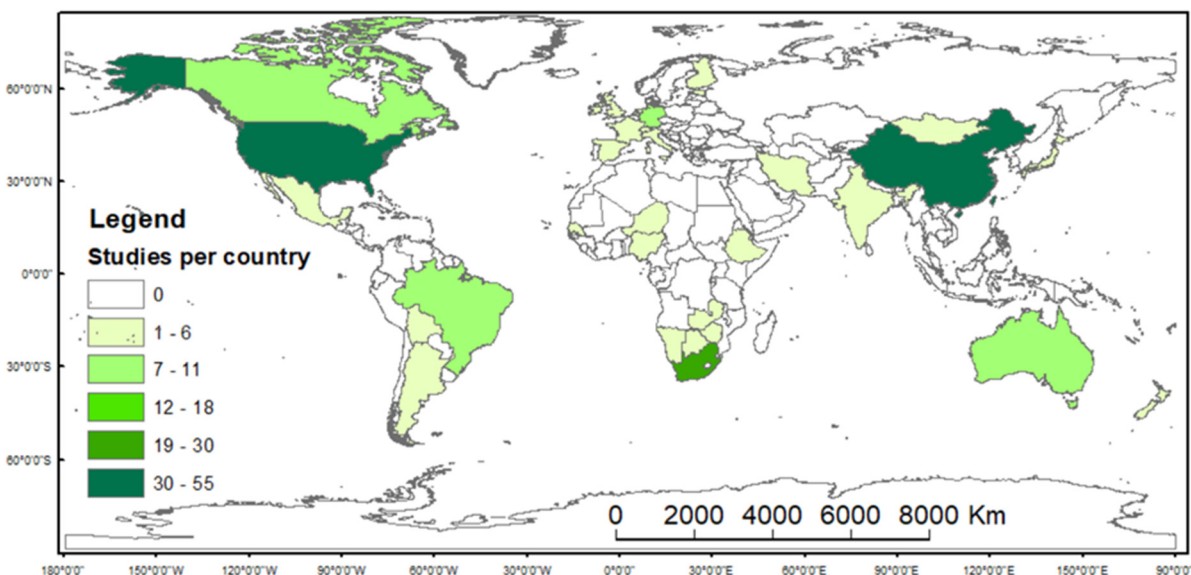

**Figure 4.** Global distribution of studies (from 1975 to 2021) that used remote sensing data to estimate grassland productivity.

Based on the articles considered in this study, it was observed that most research using RS techniques in GP was mostly conducted at the local scale (Figure 5). The least number of articles were those conducting analyses at the regional and landscape scales. While studies at the global and national scales do not demonstrate a trend, the overall number of landscape-and-regional-scale articles has gradually increased over the temporal window examined in this research.

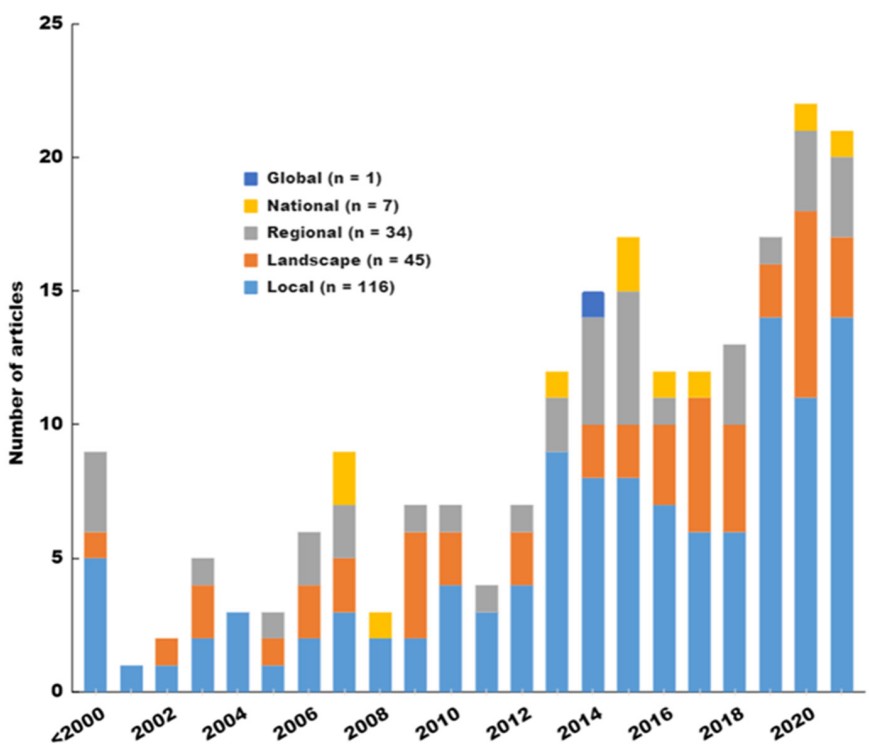

**Figure 5.** Frequency of studies conducted at different spatial scales (i.e., the extent of the study areas).

*3.4. Remote-Sensing Sensor Technologies in Mapping Grassland Productivity (Paying Particular Attention to Prediction Accuracies)*

Currently, various RS platforms with various image acquisition features are used for short-term and long-term GP monitoring. Although some studies used ground-based readings (such as the LI-COR LAI-2000 Plant Canopy Analyzer and MSR5 field-portable radiometer) and airborne sensors (such as the ASPIS sensor and HyMap hyperspectral), satellite-borne sensors account for the majority of the RS data used (Figure 6). The most widely used sensors are Landsat- 8 (OLI), AVHRR, MODIS, SPOT, and the S-2.

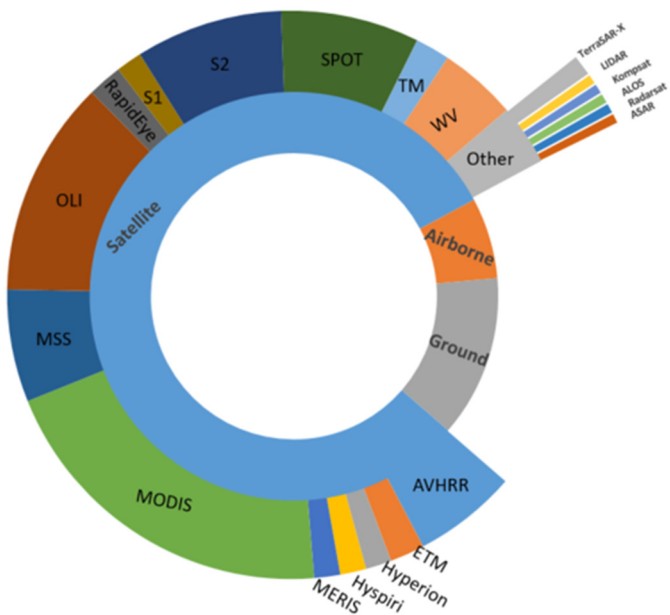

**Figure 6.** Popularly used remote-sensing sensors used in GP monitoring. WV = WorldView, S-1 = Sentinel-1, S-2 = Sentinel-2, OLI = Operational Land Imager, ETM = Enhanced Thematic Mapper, and MSS = Multi-Spectral Scanner.

The characteristics of the most frequently used satellite sensors are shown in Table 2. There is a sharp increase in the number of studies estimating GP using satellite-borne remotely sensed data. This is supported by the recent launch of new multispectral satellite sensors, such as the S-2 and Landsat-8, which can systematically acquire imagery at a high spatial resolution at no cost. However, few studies have used airborne data for estimating grassland productivity. As revealed in the literature analysis, Figure 7 shows that the few studies using radar data for grassland productivity monitoring only picked up in 2019 after the launch of S-1.

**Table 2.** Characteristics of the most frequently used sensors.

| Sensor | Bands | Spectral Range (nm) | Swath (km) | Pixel Size (m) | Temporal Resolution (Days) | Execution Scale |
|---|---|---|---|---|---|---|
| **Hyperspectral \*** | >100 | - | - | <1 | User-defined | Farm |
| **AVHRR** | 5 | 550–12,400 | 3000 | 1100 | 1 | Regional–global |
| **HyspIRI \*** | 213 8 | 380–2500 3000–12,000 | 600 150 | 60 | 19 5 | Local–regional |
| **MERIS #** | 15 | 410–900 | 1150 | 300 | 3 | Local to regional |
| **Landsat TM ETM OLI** | 7 8 11 | 450–2350 450–2350 430–12,510 | 185 | 30 | 16 | Local to regional |
| **MODIS** | 36 | 620–14,385 | 2330 | 250, 500, 1000 | 1 | Regional to global |
| **RapidEye \*** | 5 | 440–850 | 77 | 5 | 5.5 | Local |
| **Sentinel-2 MSI** | 13 | 492–1373 | 290 | 10, 20, 60 | 5, 10 | Local to regional |
| **SPOT** | 4 | 480–890 | 120 | 6,10, 20 | 26 | Local to regional |
| **SPOTVGT** | 1 | 437–1695 | 2200 | 1150 | 1 | Regional to global |
| **Worldview 2,3 \*** | 8 | 400–2245 | 16.4 | <1 | 1–1.37 | Local |
| **ALOS PALSAR \*** | VV, HH | L-band | 70 | 10 | 14 | Local |
| **Sentinel-1** | HV, VHHH, VH | C-band | 250 | 5, 20 | 6, 12 | Local to regional |
| **COSMO-SkyMed \*** | HH | X-band | ≥40 | 5 | 16 | Local |
| **TerraSAR-X \*** | VV, HH VH, HV | X-band | 270 | 1 | 2.5 | Local |

Note: Bold rows show the SAR sensors that have been used in GP. The sensors with \* and # have expensive acquisition costs, and the mission has ended.

The MODIS sensor was the most frequently used sensor, accounting for nearly 30% of all the studies, followed by Landsat 1–8 series data (23%) (Figures 6 and 7). The increasing availability of low-cost and free satellite data with moderate–coarse (100–1000 m) and moderate (10–100 m) spatial resolutions means that these are the most frequently exploited data sources for GP algorithms in the 21st century. The dominant image spatial resolutions in GP algorithms are 10 m, 30 m, and 250 m, which correlate to the S-2, Landsat series, and MODIS data. The superiority of MODIS data is explained by its frequent revisit time and large pixel size, which is computationally inexpensive for large-scale studies.

Although RS provides a profitable instrument for GP, the sensor characteristics (i.e., spectral, spatial, and temporal resolution) influence the exact retrieval of spectral reflectance observations, playing an important role in GP monitoring [38–40]. For instance, Bédard et al. [41] highlighted that different sensors exhibit unique characteristics over time and space, which affect the spectral reflectance observed by the sensor. The fine spectral resolution allows discrimination between grass and other land covers to estimate GP accurately. In addition, the phenology of the plants determines the temporal variability of their productivity [42], which can be clearly mapped by sensors with a short revisit time. Matongera et al. [43] stated that, at distinct phenological stages, grasses show variations in their productivity and water storage for the entire growing season, affecting the quantity and quality of biophysical proxies for evaluating and monitoring GP. Considering the sensor characteristics and phenology of grass species, this review found that most studies

used coarse to medium spatial resolution data with high temporal resolutions. This is supported by the studies of Xu et al. [44] and Zhang et al. [45], which highlighted that significant changes in GP need frequent, repeated monitoring to be noticeable.

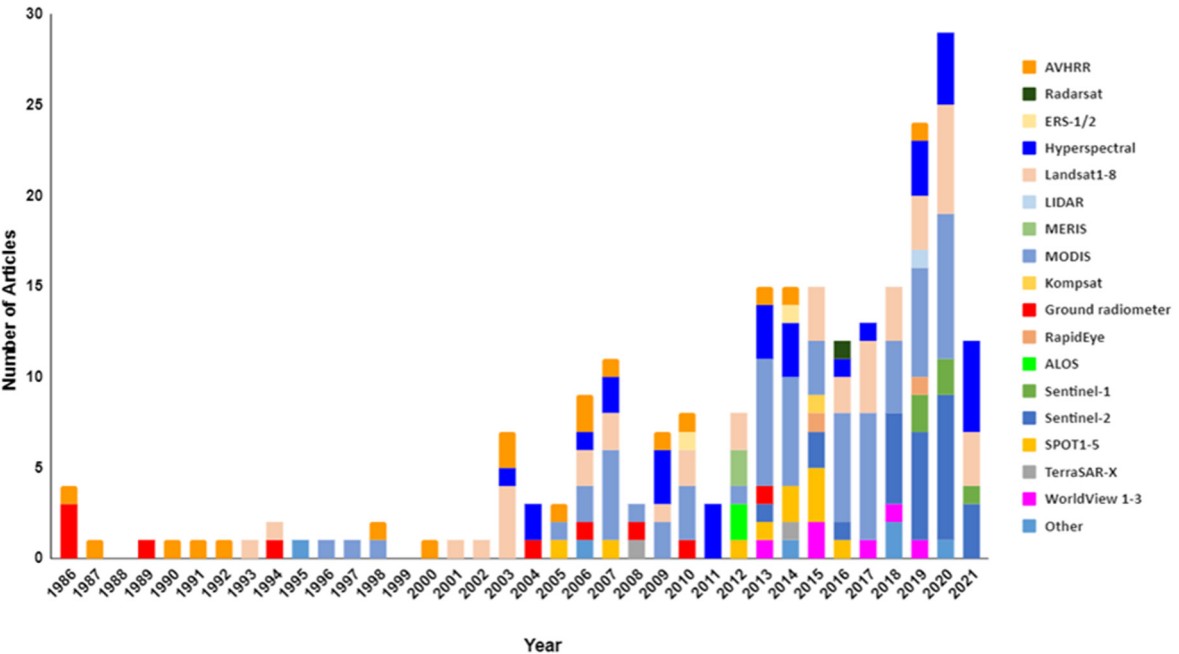

**Figure 7.** Temporal development of the remote-sensing systems used to analyze GP for the period between 1986 and the end of 2021.

Figure 8 shows the overall accuracies of the sensors that appeared in five or more studies for estimating grassland vegetation attributes. In general, the sensors show a diverse range of median overall accuracies. Hyperspectral sensors, spectra radiometers, WorldView, and S-2, have the highest median overall accuracies. The sensors with the lowest median overall accuracies include MODIS and SPOT. MODIS, Hyperspectral, S-2, and AVHRR delivered the greatest range of results, while RapidEye and WorldView provided the smallest range.

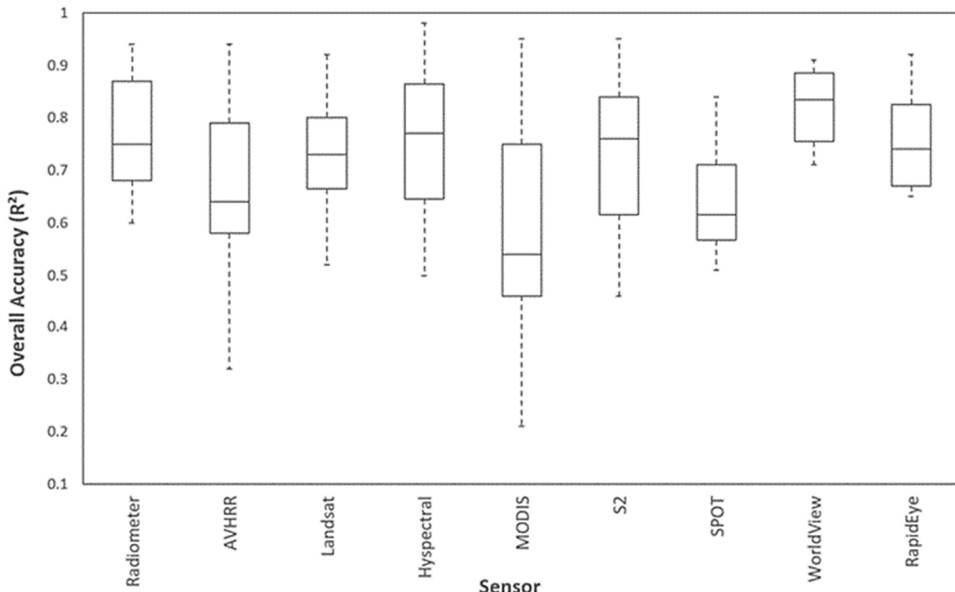

**Figure 8.** Overall accuracies of the various sensors used in monitoring grassland vegetation attributes.

The Landsat and MODIS satellites have imaged Earth's terrestrial surfaces for over forty and twenty years, respectively. This explains the extensive number of studies utilizing

these datasets in estimating the proxies of GP. Extended temporal monitoring also increases the efficacy of GP and the exploration of appropriate management practices, which increases productivity [46]. For instance, Wang et al. [47] used MODIS data to determine the annual grassland productivity of North American grassland in the period from 2000 to 2010, covering a decade, whereas Zhang et al. [48] covered a period of five (2004–2008) years.

Although coarse spatial resolution sensors have a limitation of mixed pixels, this review has found that at least 90% of the studies used data with a low spatial resolution to estimate GP over regional or global scales. Higher-resolution hyperspectral sensors yield greater precision for estimating GP by producing finer details from every pixel generated in an image. The concurrent improvements in high-resolution remotely sensed data and computer hardware and software have created an appropriate opportunity to effectively predict and map GP, regardless of the spatial and temporal extent. Thus recently, much attention has been paid to using high-resolution satellite sensors (e.g., RapidEye and WorldView-2). The study conducted by Naidoo, van Deventer, Ramoelo, Mathieu, Nondlazi, and Gangat [16] found that WorldView-3 produced the highest AGB estimation accuracies ($R^2$ = 0.65 and RMSE = 170.28 g/m$^2$) when compared to S-1 ($R^2$ = 0.56 and RMSE = 186.56) and S-2 data ($R^2$ = 0.60 and 175.08 g/m$^2$) to estimate grassland AGB.

On the other hand, Gao et al. [49] found that utilising high-resolution multispectral and hyperspectral datasets has limitations mainly coupled with the saturation of the optical signal at a high biomass density, issues in pre-processing large datasets "big data", and multi-regression and multi-collinearity problems. There are high computational costs incurred through the processing of large datasets.

Although several researchers [50–52] found strong biases caused by mixed pixels when using coarse resolution data for the estimation of GP, this review found that a few studies (about 6%) have used hyperspectral sensors (Figure 6). This is mainly because such type of data is not freely available. Thus, GP techniques are focused mostly on moderate spatial resolution satellite data, such as Landsat, SPOT, and S-2. Using such sensors produced higher predictive accuracy (Figure 8) than MODIS and AVHRR, which have a low spatial resolution.

The deployment of SAR sensors presents a state-of-the-art opportunity for retrieving biophysical parameters regardless of the weather conditions and time of day, thus holding greater potential for GP monitoring. However, few studies (<5%) have used radar data acquired by SAR sensors for GP (Figure 6). The success of SAR-based GP algorithms is reliant on the sensor (e.g., microwave frequency, polarization, and incident angle) and environmental characteristics (canopy structure, topography, soil moisture, and depth) [53].

A couple of SAR systems with different characteristics were placed in orbit during the 21st century, allowing radar RS' advancement on GP monitoring. Examples include X-band sensors, namely, TerraSAR-X with a very-small pixel size (up to 1 m) and COSMO–SkyMed, a constellation of four systems; Japanese ALOS and ALOS-2 L-band instruments; and C-band instruments on European Space Agency ASAR and S-1 sensors. Lately, systematically collected and freely available SAR datasets, such as that from the constellation of the S-1 system (Berger et al., 2012), have been unavailable. As explained in the above presentation of literature, the trends of studies based on SAR data picked up not long ago, after the launch of the S-1 satellite that provides SAR data at no cost.

To date, the RS of GP using SAR data has focused on interpreting SAR backscattering observations and the polarimetric or interferometric attributes of individual or multiple scenes to estimate vegetation parameters using linear regression models [54–57]. For example, Wang, Ge, and Li [56] analyzed the GP with COSMO-SkyMed, ASAR, and ALOS PALSAR datasets through the linear relationship between the SAR backscattering coefficient. The analysis showed that the X-band of the COSMO-SkyMed has the highest correlation ($R^2$ = 0.71) for the spatial and seasonal vegetation biomass. In another study, Wang et al. [58] estimated the seasonal dynamics of LAI and AGB using S-1, OLI, and S2 data, individually and integrally. The results from that study indicated that S-1 data

performed poorly ($R^2 = 0.44$) in recording the seasonal changes in biomass, but the accuracy improved when SAR data were integrated with optical data.

Regarding the seasonal GP using SAR data, Hajj et al. [59] observed a lower sensitivity at the X-band compared to the C-band between the microwave backscatter and the vegetation biophysical parameters with restricted possibilities of the X-band to predict GP. Inoue et al. [60] and Gao et al. [61] undertook studies to compare the sensitivity of individual portions of the electromagnetic spectrum to plant biomass, while Barrett et al. [62] demonstrated how the most accurate monitoring of grasslands could be attained with the integrated use of L- and C-bands time-series. In addition to using multiple wavelengths, different polarimetric acquisition capabilities can be exploited. For instance, Pairman et al. [63] utilized TerraSAR-X dual polarimetric SAR multi-images to estimate pasture biomass. In addition, using the multiple angles of Radarsat-2 quad-polarization, Buckley and Smith [64] monitored GP for prairie grasslands. Their comparison with a single incidence angle showed better results for grassland classification.

While SAR data have many advantages, such as the ability to observe the ground even during cloudy conditions, there are also challenges related to SAR image analysis and its interpretation for estimating biophysical parameters for GP monitoring [65]. A further limitation associated with SAR imagery is the presence of speckles, which can affect observation accuracy and therefore cause a decrease in classification accuracy [65]. The backscatter signal from grasslands is governed by several variables, including the radar system, vegetation (i.e., stems and leaves), soil surface, and the biophysical parameters of the scatterers in the grass water content. Wang, Ge, and Li [56] concluded that the accuracy of SAR data in GP can be influenced by rain due to its sensitivity to water drops on leaves. To date, SAR data use in GP has remained rudimentary, except for a few studies [66]. Therefore, a few (<5%) studies have used these sensors. However, recently, Wang, Xiao, Bajgain, Starks, Steiner, Doughty, and Chang [58] found the best performance from the integration of LS8, S1, and S2 ($R^2 = 0.76$) than the S-1 backscatter signals ($R^2 = 0.04$–$0.44$).

Recently, the performance of light detection and ranging (LiDAR) technique was reported in estimating grassland biophysical parameters using RS data [67,68]. Meanwhile, Zhang, Bao, Wang, Xin, Ding, Xu, Hou, and Shen [67] found that the ground-measured biomass was correlated ($R^2 = 0.54$) with LiDAR estimates.

### 3.5. Utility of Vegetation Indices as Proxy for Estimating Grassland Productivity

An important and widely studied method of GP prediction is the use of RS-derived vegetation indices (VI) in conjunction with in situ observations [69–71]. Most VIs use the red, red-edge, and near-infrared wavelengths [15,70,72]. Over 80% of the studies tested the utility of NDVI in estimating grassland productivity. NDVI is influenced by the soil background signature and is associated with saturation problems at high LAI [73–75]. This has meant that more studies have incorporated the modified version of NDVI and other red-edge-based vegetation indices to address saturation issues [73,76,77]. According to Mutanga and Skidmore [73], the integration of narrow bands in the shorter wavelengths of the red edge (700–750 nm) and longer wavelengths of the red edge (750–780 nm) produce higher correlations (average $R^2 = 0.77$ for the top 20 NDVI values) with grass biomass than the standard NDVI alone.

The second most used VI for estimating grassland productivity is the enhanced vegetation index (EVI), which was formulated to improve biomass estimation in areas with elevated biomass by eliminating the canopy background signal and depletion in atmospheric influences [78–80]. EVI uses the blue band, which is absent in most sensors. This drawback has resulted in the development of the two-band $EVI_2$ for sensors such as the AVHRR, which have a missing blue band [81]. The articles by Kim et al. [82] and Jarchow et al. [83] reported the similar performance of the EVI and $EVI_2$ bands when estimating grassland productivity at large scales. The earlier study's accuracy was very high for EVI ($R^2 = 0.97$) and $EVI_2$ ($R^2 = 0.98$). Therefore, $EVI_2$ can provide the continuity of the monitoring of grassland productivity across sensors with different spectral bands.

Transformed normalized difference vegetation index (TNDVI), perpendicular vegetation index (PVI), soil-adjusted vegetation index (SAVI), modified soil-adjusted vegetation index (MSAVI), transformed soil atmospherically resistant vegetation index (TSARVI), and transformed soil-adjusted vegetation index (TSAVI) is another group of modified VIs used in GP monitoring to minimize the effects of the soil on the vegetation spectral profile. The evidence from the literature has shown that the applications of this group of VIs in GP are primarily appropriate for regions with reduced grass cover, where the influence of soil brightness is high [75,84,85]. For instance, Ullah, Si, Schlerf, Skidmore, Shafique, and Iqbal [85] examined natural GP in the northern Netherlands, characterized by short grass cover, and found that SAVI ($R^2$ = 0.54), TSAVI ($R^2$ = 0.52), and NDVI ($R^2$ = 0.51) have similar performances. In another interesting study conducted by Jin et al. [86] over a large landscape (193,000 $km^2$), the statistical analysis showed that the $R^2$ of NDVI, SAVI, and MSAVI were 0.686, 0.702, and 0.69, respectively.

The renormalized difference vegetation index (RDVI) is another VI widely used for GP [75,87,88]. RDVI combines the advantages of difference vegetation index (DVI) and NDVI for low and high vegetation cover, respectively. The literature shows that RDVI is sensitive to LAI changes, making it suitable for productivity monitoring in grasslands with a high LAI [77]. For example, Tagesson et al. [89] found that RDVI ($R^2$ = 0.90) performed better than NDVI ($R^2$ = 0.79) in areas with a high LAI.

VIs based on the red edge (680–780 nm) have been proposed to minimize the impacts of the bidirectional reflectance distribution function and background noise, resulting in a better performance when estimating grassland productivity [38,90,91]. For instance, Lin, Li, Liu, Li, Zhao, and Yu [90] evaluated the efficacy of using VIs based on red-edge reflectance from S2 over a small local area to estimate grassland productivity. The results showed a high correlation ($R^2$ = 0.77). Recently, Imran et al. [92] observed a strong correlation ($R^2$ > 0.8) between VIs based on the red edge and grassland LAI in a local grassland using S2 data. The major problem in using the VIs obtained from red bands is the small sensitivity to over-story vegetation conditions [93].

Lately, advances in hyperspectral and commercial multispectral sensors have facilitated the evolution of narrowband greenness VIs, which have become the most suitable approach for assessing GP [16,94]. These narrowband indices can successfully discriminate discrete grass biochemical properties, such as chlorophyll.

Although VI remains the most used GP indicator, some vegetation biophysical parameters, such as LAI [58,95,96], the fraction of absorbed photosynthetically active radiation (FAPAR) [90], canopy storage capacity [97], canopy chlorophyll content [95,98], and the greenness factor, have also been used to estimate productivity in grassland ecosystems. The evidence from the literature shows that these biophysical parameters provide detailed information about the grassland's physiological health, which is a very important indicator of photosynthetic potential [95,97].

Although VI regression models have been reported in the literature to have very-high accuracies in estimating biomass [99,100], their major limitation is that these models are site-specific and cannot deal with highly non-linear and complex patterns in the data. A major challenge in using VIs to assess GP is to minimize the influence of external factors and maximize the sensitivity of the relationship between VIs and biophysical parameters.

### 3.6. Algorithms Used for Grassland Productivity Using Remote Sensing

This study noted that the RS of GP can be performed using either physical-based (i.e., radiative transfer models (RTMs)) or empirical/statistical models (Table 3). Although 1-D and 3-D RTM inversion approaches have proven to be a promising way to retrieve the biophysical and biochemical variables of proxies of vegetation, such as the leaf area index (LAI), canopy water content, canopy or leaf chlorophyll content, and fuel moisture content [17,95], this review found that only 6% of the studies used this technique for GP. RTM-based approaches have the benefit of reproducibility. These models are more general and are based on physical laws that establish explicit relationships between canopy

properties and spectra. Outstanding results, with an $R^2$ greater than 0.7, were reported for using RTMs [101,102]. For example, Quan, He, Yebra, Yin, Liao, Zhang, and Li [17] used the PROSAILH (PROSPECT + SAILH) model and reflectance from OLI product to derive LAI and AGB in a grassland wetland. The RTM-based approach yielded a higher accuracy ($R^2 = 0.64$) than the exponential regression ($R^2 = 0.48$) and the ANN ($R^2 = 0.43$). Another example of a successful ($R^2 > 0.7$) RTM-based estimation of GP is the function of the crop growth model by Bella, Faivre, Ruget, Seguin, Guerif, Combal, Weiss, and Rebella [100] in France. RTM approaches, on the other hand, are computationally demanding, particularly when complex models are used. This makes retrieving biophysical variables over vast geographic areas impossible [95,101].

NDVI and in situ measurements of the empirical/statistical algorithms used to obtain a correlation between the spectral data or VIs with biomass are the widely used approaches in GP [42,103]. The empirical approach relates RS variables to in situ grass biomass via parametric or non-parametric regression models [42]. Linear regression, which has been used for decades, is the widely used algorithm for GP because of its simplicity and computational efficiency. However, the derived statistical relationships ($R^2$ ranging from 0.25 to 0.70) are considered sensor-specific, site- and sampling-conditions-dependent, and are anticipated to change in space and time [42]. Using MODIS-derived grass biomass, Xu et al. [44] compared three different regression models, and the best correlation was shown by an exponential algorithm ($R^2 = 0.80$), followed by the PLSR ($R^2 = 0.79$) and linear ($R^2 = 0.67$).

**Table 3.** Available algorithms for grass productivity prediction using remotely sensed data.

| Algorithm | Remote-Sensing Datasets | Performance | GP Parameter(s) | References |
|---|---|---|---|---|
| Linear regression | MODIS | $R^2$ varied between 0.25 and 0.68. | AGB | [104] |
| | AVHRR | $R^2$ ranged from 0.39 to 0.47. | AGB | [105] |
| | MERIS | $R^2$ ranged from 0.51 to 0.72. | Nitrogen and AGB | [85] |
| Exponential regression | Landsat 8 OLI | The RTM-based algorithm yielded higher prediction values ($R^2 = 0.64$) than the exponential regression ($R^2 = 0.48$) and ANN ($R^2 = 0.43$). | LAI, leaf chlorophyll content, leaf water content, and AGB | [17] |
| PLSR | | | | |
| PROSAILH | | | | |
| SML | Sentinel-2 | The RMSE was 10.86 g/m$^2$, and the $R^2$ accuracy was 82.84%. | AGB | [88] |
| SPLSR | Sentinel-2 and HyspIRI | HyspIRI data showed higher AGB prediction accuracies (RMSE = 6.65 g/m$^2$, $R^2 = 0.69$) than those from S-2 (RMSE = 6.79 g/m$^2$, $R^2 = 0.58$). | AGB | [106] |
| PLSR | Hyperspectral | Results showed that PLSR models could retrieve LAI on hyperspectral images with accuracy values ranging from 0.81 to 0.93. | LAI | [107] |
| RF | WorldView-2 | Results showed that random forest and vegetation indices achieved >89%. | Leaf nitrogen and AGB | [18] |
| | S-2 and OLI | $R^2$ ranges from 0.84 to 0.87. | LAI | [108] |
| SVM | Radarsat-2 | The SVM yielded the best overall prediction ($R^2 = 0.98$) for GP in central-north Brittany, France. | LAI | [109] |
| | MODIS | SVM ($R^2 = 0.58$ and RMSE = 5.6 g/m$^2$). | AGB | [110] |
| | Hyperspectral | SVM models yielded higher accuracies ($R^2 = 0.90$) than PLSR models ($R^2 = 0.87$). | LAI | [96] |

**Table 3.** *Cont.*

| Algorithm | Remote-Sensing Datasets | Performance | GP Parameter(s) | References |
|---|---|---|---|---|
| ANN | Landsat 7 ETM+ | The study showed the AGB values modeled by ANN ($R^2 = 0.817$) were not far from the observed values than MLR ($R^2 = 0.591$). | AGB | [111] |
| DT | ENVISAT ASAR, ERS-2 | Overall accuracies $R^2 \geq 88.7\%$ were achieved for most datasets. | AGB | [62] |
| PROSAIL | S-2 | The $R^2$ ranged from 0.22 to 0.76. | LAI, AGB, and leaf chlorophyll and water content | [112] |

While parametric models can yield moderate estimates for GP, they are associated with several challenges. For instance, Verrelst et al. [113] indicated that parametric algorithms lack proficiency in expressing the complex relationships between RS variables and grass aboveground biomass. Furthermore, researchers have noted that parametric methods suffer from multi-collinearity, overfitting, and produce unstable predictions, when working with small sample sizes and missing values [107,114].

Conversely, the new-fashioned and resilient non-parametric MLAs, including support vector machines (SVM), random forest (RF), partial least squares regression (PLSR), sparse PLSR (SPLSR), boosted regression trees (BRT), and artificial neural network (ANN), presents a robust tool for estimating GP. MLAs have been reported to be powerful, efficient, and less affected by the dimensionality of data than parametric algorithms [18,115]. Furthermore, non-parametric algorithms overcome the challenges associated with using parametric algorithms, such as multi-collinearity, overfitting, handling small sample sizes, and missing values [42,116]. Though these algorithms are increasingly replacing parametric algorithms for GP, they are still considered rudimentary in the domain of GP [48,49,96].

However, some inevitable limitations are associated with using MLAs for GP. The accuracy of the results is strongly determined by the quality of the training dataset. The existence of outliers and erroneous values in the training data may weaken the model performance [42]. Some MLAs, such as ANNs, are complex, computationally demanding, and require adjustments of several parametrizations, such as the kennel size [117]. In addition, some MLAs tend to be suitable for specific locations; hence, the models developed are not adapted to other environments, and in the case of RF, it has been documented that the algorithm tends to underestimate the high values and exaggerate the low values of AGB [42]. However, these issues are notable, and researchers try to minimize these limitations with precise strategies. For instance, the availability of vast datasets will aid remotely sensed data-driven models to achieve better generalization.

## 4. Discussion

### 4.1. Algorithms Used for Grassland Productivity Using Remote Sensing

As discussed in the previous sections, GP monitoring has been widely studied in the last five decades. However, many studies have been conducted in Asia, and few attempts have been made to estimate grassland AGB using RS in African and Australian grasslands, especially on indigenous grasses [16,118]. The limited number of studies in estimating grassland productivity using RS in some countries undermines the appreciation of grasslands in the carbon cycle.

Currently, there is a great demand for GP information at larger scales. Therefore, the future of regional studies for GP is invested in using RS datasets with applicable pixel sizes, spectral resolution, and repeat cycles in conjunction with algorithms, which can improve performance accuracy. The progress in satellite RS and using UAVs for collecting images at fine resolutions have revived GP monitoring [80]. Grasslands play a significant role in carbon sequestration and support ecosystem services; therefore, the use of RS in

GP monitoring will be uninterruptedly enhanced in the next decades as climate change mitigations increase.

Consequently, the availability of affordable and advanced sensors with a fine pixel size (e.g., RapidEye and WorldView 3), quick revisit time (e.g., Hyperspectral InfraRed Imager (HyspIRI) and the constellation of S-1A and 1B, S-2A and 2B, and S-3), fine spectral resolution (e.g., S-2 MSI), and radiometric resolution (e.g., Landsat 8 OLI) creates advanced opportunities for GP monitoring. We draw attention to the newly launched HyspiIRI sensor, which has 213 spectral bands between 380 and 2500 nm, aiding the observation and characterization of exquisite contrast in grass species that are unnoticeable using broadband multispectral sensors [42,43]. In addition, the OLI has finer spectral bands, sophisticated calibration and signal-to-noise characteristics, higher 12-bit radiometric resolution, and more advanced geometry compared to its predecessors [119]. Therefore, these sensors provide greater possibilities for future studies in quantifying grassland quantity parameters (i.e., biomass, LAI, and CSC).

Studies that have used datasets acquired by new-generation sensors have shown that they have great potential to predict GP. For instance, Sibanda et al. [120] and Naidoo, van Deventer, Ramoelo, Mathieu, Nondlazi, and Gangat [16] discovered that the WorldView-3 sensor has a special red-edge band, which has a high potential to estimate grassland productivity. In a related study, Sibanda, Mutanga, and Rouget [106] reported that the hyperspectral resolution of the HyspIRI imagery has a high potential to monitor grassland productivity, especially in heterogeneous environments. Given this, future research on the GP is enlightened. The advanced properties of new-generation datasets are more likely to offer an improved temporal characterization of grasslands to effectively manage grasslands and maintain ecosystem services.

Studies that have used SAR datasets have proved they have the potential for estimating proxies of GP, especially in areas where cloud cover is very high most of the time, which limits the use of optical satellite data [57,109]. Dusseux, Corpetti, Hubert-Moy, and Corgne [109] found that the classification accuracy of SAR variables is significantly higher than those using optical data (0.98 compared to 0.81). The authors highlighted that integrating optical and SAR remotely sensed data is of prime interest in distinguishing grass classes from other features.

The use of advanced MLAs in GP has also been successful when compared to conventional algorithms, even when using broadband multispectral sensors. Advanced MLAs improve the use of RS data to quantify GP [114,121]. Generally, studies conducted in grassland ecosystems have reported the potential of MLAs in estimating GP [18,122].

When satellite remotely sensed datasets are absent, it is also possible to collect remotely sensed data using UAVs to explore the potential of remotely sensed data in predicting GP [80]. The resampling of hyperspectral images collected from UAVs is becoming a reliable alternative in testing the potential of available or upcoming sensors' spectral configurations, especially considering the limitations linked with hyperspectral datasets.

This study revealed that most research on estimating vegetation chlorophyll at the leaf and canopy scale has been for precision agriculture or forests, and few have been conducted for grassland ecosystems. There is a need to improve the generality and applicability of VIs for estimating GP at the leaf level. Studies at this scale can reveal key information applicable to ecosystem health, such as grass' physiological status, productivity, or phenology.

*4.2. State-of-the-Art Approaches for Improving GP Monitoring Using Remote-Sensing Techniques*

While great progress has been made in sensor development and GP monitoring approaches, several important issues for improving estimating GP—especially in complex environments (e.g., woody grasslands and grasslands with mixed grass species) such as in many places—need to be paid more attention to. Firstly, notwithstanding the extensive research on land use and land cover classification, few studies have focused on solving the issue of mixed pixels, which is relevant to the mapping of GP using coarse-spatial, high-temporal resolution imagery. There is a cut-off between studies on RS for GP in hetero-

geneous environments and the operationalization of remotely sensed data for grasslands application. A transformation from science-driven techniques to explicit, user-oriented approaches of RS is required for monitoring grasslands and the dynamics of grasses in heterogeneous environments.

More research is required to assess the prospects of the SAR and hyperspectral datasets, specifically those from S1 satellite and airborne (e.g., unmanned aerial vehicles (UAVs)) sensors. S-1 data sources hold much potential for GP as they are freely available, weather- and daylight-independent radar systems, have relatively high spatial resolutions and have short revisit periods. In addition, regarding the poor economic states of most countries, GP-monitoring techniques should be robust, cheap, and autonomous. When it comes to GP at the farm scale, the use of UAVs proved to be useful. Farmers can plan their pasture management methods and grazing capacity accordingly. This economically viable and easy technique will greatly improve the sustainable management of grasslands. Eventually, the compilation of robust and fruitful local-to-regional frameworks and policies to ease sustainable grassland management practices are more likely to be accomplished.

The emergence of remotely sensed datasets with a high temporal resolution has paved the way for near-real-time GP monitoring. Every day, a large number of space-borne and airborne sensors provide a considerable amount of remotely sensed data. These data are becoming an economic asset and a new important resource in estimating grassland productivity. There is, thus, a need to develop powerful data analysis techniques, such as MLAs, data fusion, multi-sensor approaches, and cloud-based storage and processing systems (e.g., Google Earth Engine) to handle large datasets.

RTM inversion algorithms have been demonstrated to be a promising way to retrieve proxies of GP. These models are more adaptable as they are based on physical laws that provide straightforward connections between canopy and LAI properties and spectra [17]. Furthermore, these RTM-based approaches have the advantages of replicability and robustness at a large scale without the need to gather field samples. Consequently, there is a need to channel research towards using RTMs in estimating GP, especially in large areas. The recent study by Berger et al. [123] proved that RTMs are computationally expensive and require high-end computers to perform quickly. This is no longer a drawback since computer technology is advancing rapidly, particularly with the advent of cloud computing platforms such as Google Earth Engine.

### 4.3. Limitations and Future Expectations on Applications and Sensors

Given the spatial and spectral variability of grass species, finding the correct dataset, with the optimal spectral and spatial resolution, remains a crucial drawback for estimating GP using RS data. Currently, GP estimates in light of climate change are required at the regional or national scale. These sensors must have the optimum spectral and spatial resolution sufficient to provide very-high spatial resolution data. However, the spatial resolution of current multispectral data products acquired in wide swaths have a low temporal resolution (10 days at the equator with one satellite), negatively affecting the performance of techniques for explicitly estimating GP.

The available data on predicting GP over large areas have been derived using coarse spatial resolution datasets, such as MODIS (Figure 5). Despite having global coverage, a short revisit time, and being in operation for more than five decades, which are prerequisites for prolonged monitoring, MODIS data have a low prediction accuracy (see Figure 8 and Table 2), especially in heterogeneous grasslands. The limitations of the MODIS pixel size are inappropriate to adequately estimate GP in mixed grassland ecosystems [83]. Although the coarse resolution sensors are associated with vast errors in the estimation of AGB, LAI, and CSC, they are still widely used for large-scale GP monitoring.

Fine spectral and spatial resolution input data are crucial for GP estimation approaches. To date, space-borne hyperspectral data have not been easily accessible, constraining studies to the local scale using UAV data. Studies using broadband multispectral data have produced low prediction accuracies in contrast with hyperspectral datasets, for example,

the research conducted by Jarchow, Didan, Barreto-Muñoz, Nagler, and Glenn [83]. In this regard, the trade-offs between pixel sizes, image acquisition costs, swath width, and spectral resolution were reported by many authors as a major drawback of RS for estimating GP and their function in the C cycle [40,42,124].

This review study showed several proposed RS techniques for GP monitoring. To date, there is not much information about which algorithm is superior in performance, because a limited number of the reported approaches have been validated for their accuracy. To date, few comparative studies have been undertaken. In addition, although VIs are successful in GP monitoring, it is not suitable in heterogeneous grasslands, particularly when scattered trees or shrubs are present. These indices are not robust and are site-specific. The conventional NDVI, developed from the red and NIR bands, have been reported to perform poorly in sparsely vegetated areas and associated with saturation challenges in thickly vegetated areas [73,88]. This has led to the development of other indices with better accuracies than the NDVI, such as the red-edge NDVI (NDVI derived using red edge bands), EVI, SAVI, and MSAVI. Using the MODIS data, Zhou et al. [125] found that MSAVI, EVI, and SAVI indices outperformed the NDVI in quantifying the productivity of northern China grasslands.

There is a need to adopt novel methods to enhance the accuracy of estimating proxies of GP. Deep learning has been widely used in many areas of image processing due to its effective performance in processing high-dimensional data and non-linear relationships. Estimating GP proxies using time series of hyperspectral imageries over large areas takes significant effort using machine-learning regression approaches. This means that deep learning is an outstanding alternative. Deep learning can also deal with multi-source inputs and learn weights to combine important information from them.

Despite the greater use of remotely sensed data in grassland monitoring, fewer studies have combined the several features of GP to develop an exhaustive, consistent biophysical monitoring system. In order to improve the eligibility and transferability of biophysical simulation models, it is also necessary to integrate multiple-source remote-sensing data.

## 5. Conclusions

The progress of RS systems and the introduction of new advanced classification algorithms have gained the attention of researchers to use these potential data and tools for GP monitoring. RS is fully used to obtain accurate information on GP proxies: AGB, LAI, CSC, and chlorophyll content. The present study analyzed a substantial body of literature by gathering a comprehensive dataset (203 articles) for estimating grassland productivity using RS until the end of 2021.

The number of studies on estimating GP using remotely sensed data has risen significantly, but most strongly in the last two decades. The prevalence of research on GP is unequally distributed globally, as China has the highest number of studies, followed by the USA. Few studies have been conducted in Africa and Europe.

Multispectral satellite data was used in 88% of studies, especially in studies focusing on the retrieval of AGB. Few studies (10%) used microwave systems, and only 2% combined optical and radar data. Although hyperspectral data are associated with better accuracy than multispectral data, they have a small swath width, high acquisition costs, and high pre-processing. These challenges have limited their use, enabling researchers to focus more extensively on unrestrained broadband multispectral datasets.

The progress in techniques can further enhance the accuracy in estimating GP using the optimal RS datasets. The selection of the most appropriate image classification algorithm is one of the current topics of discussion in GP monitoring using RS data. Thus, there is a call for future research to test the applicability of broadband multispectral and hyperspectral sensors with state-of-the-art image acquisition traits, combined with powerful MLAs, such as DA, RF, PLSR, SVM, and ANN, for the well-informed management of grasslands.

**Author Contributions:** Conceptualization, T.B., O.M., T.D., M.S. and T.M.; methodology, T.B., O.M., T.D., M.S. and T.M.; writing—original draft preparation, T.B.; writing—review and editing, O.M.,

T.D., M.S. and T.M.; funding acquisition, O.M. and T.M. All authors have read and agreed to the published version of the manuscript.

**Funding:** This research was funded by the Water Research Commission of South Africa, Project number WRC2020/2021-00490 titled "Geospatial modelling of rangelands productivity in water limited environments of South Africa".

**Data Availability Statement:** Data are available upon request.

**Conflicts of Interest:** The authors declare no conflict of interest.

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
