# Peer review of "Remote Sensing Grassland Productivity Attributes: A Systematic Review"

_remotesensing, doi:10.3390/rs15082043_

Round 1
Reviewer 1 Report
This manuscript presented a comprehensive review of remote-sensing-based grassland productivity assessment, including the dataset, publication trends, algorithms, uncertainties, challenges, and future potential ways to improve grassland productivity estimation. The ms is well organized following the typical structure of a review paper. My recommendations for the authors are, in the discussion section, to pay attention to Radiative Transfer Models, there are 3-D RTMs available to monitor the vegetation. Another is the data-driven deep learning algorithms since it already dominates the computer vision domain, and should be a must-known-mastered technics for grassland monitoring. MORE, the manuscript needs proofreading, there are a lot of grammar errors and prompts showing errors/reference sources not found.
Author Response
Dear Reviewer,
Please see the attachment.
Regards,
Tsitsi

Reviewer 2 Report
This review has a rigorous structure, clear methodology, and comprehensive data, comprehensively elaborating the achievements of existing research, which provides important reference value for subsequent relevant studies.
There are several mistakes:
(1) Line97: three stages or four stages? Please check.
(2)line112,123,180.... Error! Reference source not found. please check(3)figure 6.................bold
Author Response
Dear Reviewer,
Thank you very much for taking your time to review our paper. Please see the attachment for the feedback.
Regards,
Tsitsi

Reviewer 3 Report
Please find the attached file for my comments.

Author Response
Dear Reviewer,
Thank you very much for taking your time to review our paper. We appreciate all the time you spent reading our paper. Please the attachment.
Regards,
Tsitsi

Reviewer 4 Report
Dear [Authors],
I have reviewed your paper titled "Remote sensing of grassland productivity monitoring: A systematic review" and I found the paper to be well-organized and analyzed, and I appreciate the effort you put into your research.
However, I would like to suggest some minor revisions to the paper. Firstly, I noticed some English spell check issues that need to be addressed to ensure the clarity and coherence of the paper. I recommend that you review the text carefully.
Secondly, I suggest that you improve the quality of the figures in the paper. While the figures are useful in supporting the analysis, they could benefit from clearer labeling and higher resolution images. Also check for typos and fonts (line 149-183)
Lastly, there are some references missing from the paper which makes it difficult to review. Line 112-123-180-189.
In conclusion, I recommend you that make the suggested revisions to improve the quality of the paper and make it more accessible to readers.
Thank you for your hard work and contribution to the Remote Sensing journal.
Sincerely,
Author Response
Dear Reviewer,
Thank you very much for taking your time to review our paper. The suggestions were constructive and have enlightened our research career. Please see the attachment.
Regards,
Tsitsi
